# Risk of metabolic syndrome in patients with lichen planus: A systematic review and meta-analysis

**Jieya Ying[1], Wenzhong Xiang[2]\*, Yu Qiu[1], Xiaofang Zeng[1]**

**1** Fourth Clinical Medical College, Zhejiang Chinese Medical University, Hangzhou, China, **2** Department of Dermatology, Hangzhou Third People's Hospital, Hangzhou, China

\* xiangwenzhong@126.com

## Abstract

### Background

Studies have investigated whether patients with lichen planus are at a high risk of metabolic syndrome; however, currently, no conclusive data are available in this regard.

### Objective

This meta-analysis was performed to analyze the published literature investigating the association between metabolic syndrome and lichen planus.

### Method

Two reviewers independently searched 4 databases (PubMed, Embase, the Cochrane Library and Web of Science) for observational studies assessing the prevalence of metabolic syndrome in patients with lichen planus. Review Manager 5.3 software was used to statistically analyze the data.

### Results

200 relevant articles were searched. After a further reading, 12 studies with 1422 participants (715 with LP and 707 controls) fulfilled the eligibility criteria. Overall, the pooled odds ratio based on random effects analysis was 2.81 (95% confidence interval: 1.79–4.41, $P<0.00001$). This meta-analysis shows that compared with the general population, patients with lichen planus are more likely to develop metabolic syndrome. Subgroup analysis of prevalence of metabolic syndrome showed higher odds ratio in studies using International Diabetes Federation diagnostic criteria (odds ratio 4.65) and the Harmonized criteria (odds ratio 26.62) than studies using National Cholesterol Education Program Adult Treatment Panel III criteria (odds ratio 1.75), and thus might be more appropriate for diagnosing metabolic syndrome.

**Data Availability Statement:** All relevant data are within the paper and its Supporting Information files.

**Funding:** WX received a research grant (No. 2020364003) from Zhejiang Medical and Health Science and Technology Program, China (URL: http://www.msttp.com/). The funder had no role in study design, data collection and analysis, decision to publish, or preparation of the manuscript.

**Competing interests:** The authors have declared that no competing interests exist.

## Conclusions

This meta-analysis shows that compared with the general population, patients with lichen planus are more likely to develop metabolic syndrome. Therefore, early diagnosis and prompt initiation of first-line therapy for metabolic disorders are important in patients with lichen planus.

## Introduction

Lichen planus (LP) is an idiopathic inflammatory dermatosis of the skin and mucous membranes characterized by flat, pruritic, violaceous polygonal papules. The overall prevalence of LP is approximately 0.89% of the general population [1], and it commonly affects individuals aged >45 years [2]. Although the etiopathogenesis remains unclear, LP is attributed to a T cell-mediated autoimmune process that results in degeneration and destruction of keratinocytes [3]. Recent studies have reported that in addition to affecting the skin, LP may be associated with metabolic syndrome (MS).

MS is a complex group of metabolic disorders, which include central obesity, dyslipidemia, hypertension, and hyperglycemia with insulin resistance as the central pathophysiological feature. It was first described as syndrome X by GM Reaven in 1988 [4]. The other terms, such as Reaven syndrome and the insulin resistance syndrome, have also been largely used in the literature for several decades [5].

MS is implicated as an important risk factor for type 2 diabetes mellitus and cardiovascular disease and has therefore attracted increasing attention in recent years. Growing evidence has shown that some dermatological diseases such as psoriasis [6] alopecia areata [7], hidradenitis suppurativa [8], atopic dermatitis [9] and vitiligo [10] are associated with a high prevalence of MS compared with the general population. LP and psoriasis show a similar pathological background, including features of skin barrier dysfunction [11], T lymphocyte activation, and upregulation of cytokines such as tumor necrosis factor (TNF)-α, interleukin (IL)-6, IL-10, and IL-4 [12] and may therefore be associated with a high risk of MS. It has been confirmed that LP is associated with dyslipidemia, indicated by increased serum triglyceride (TG), cholesterol, and low-density lipoprotein cholesterol (LDL-C) levels and decreased high-density lipoprotein (HDL-C) levels [13].

Studies have also investigated whether patients with LP are at a high risk of MS; however, currently, no conclusive data are available in this regard. Therefore, this meta-analysis was performed to analyze the published literature investigating the association between MS and LP and determine whether patients with LP are more likely to develop MS when compared with the general population.

## Materials and methods

### Data sources and searches

In this systematic review and meta-analysis, two reviewers (JY and YQ) independently searched the literatures in accordance with the Preferred Reporting Items for Systematic Reviews and Meta-Analyses (PRISMA) guidelines (see S1 Checklist for complete PRISMA checklist). We searched PubMed, Embase, the Cochrane Library and Web of Science for observational studies assessing the prevalence of MS in patients with LP published from their inception to July 16, 2020. To search relevant studies, the following terms were used:

"(metabolic syndrome OR syndrome X OR insulin resistance syndrome OR Reaven syndrome) AND (lichen planus OR lesion planus)" (see S1 Table).

## Inclusion criteria

The inclusion criteria for this study were as follows: (1) For studies: only studies reported in English, and observational study design (cross-sectional, cohort and case-control). (2) For experiment groups: LP affecting the skin or mucous membranes, no systemic treatment administered, and no topical therapy administered for >6 weeks. (3) The control groups included individuals without LP, who were randomly selected from the communities or hospitals. (4) Outcome measures: MS was diagnosed using the National Cholesterol Education Program Adult Treatment Panel III criteria (NCEP ATP III) (published in 2001) [14], the International Diabetes Federation (IDF) criteria (published in 2006) [15], the Harmonized criteria (proposed by IDF, National Heart, Lung and Blood Institute, American Heart Association, World Heart Federation, International Atherosclerosis Society and International Association for the Study of Obesity in 2009) [16] and other diagnostic criteria (Table 1). LP was diagnosed based on clinical manifestations and histopathological examination of biopsy specimens. Sex, race, and publication year were not considered in study selection.

## Exclusion criteria

The exclusion criteria for this study were as follows: (1) For studies: study types including reviews, comments, guidelines, case reports, letters, conference abstracts, or laboratory research. (2) For all participants: a known diagnosis of oral lichenoid reactions (a drug-induced LP-like reaction), hypertension, diabetes, dyslipidemia, chronic liver disease, chronic kidney disease, human immunodeficiency virus infection, and hereditary diseases; patients with psoriasis, atopic dermatitis, vitiligo or other diseases that are associated with MS. (3) incomplete or incorrect data.

## Data collection

The retrieved literatures were screened for relevance in the titles and abstracts to determine whether they fulfilled the study selection criteria. Two authors (YJ and XZ) independently

**Table 1. NCEP ATPIII, IDF and Harmonized criteria for MS.**

| | NCEP ATPIII [14] | IDF [15] | Harmonized criteria [16] |
|---|---|---|---|
| **Waist circumference expanding** | >102 cm in males and >88 cm in females | ≥94 cm in males and ≥80 cm in females | ≥94 cm in males and ≥80 cm in females |
| **Hyper-triglyceridemia** | ≥150 mg/dL | ≥150mg/dl or specific treatment for this lipid abnormality. | ≥150mg/dl or specific treatment for this lipid abnormality. |
| **Low HDL-C** | <40 mg/dL in males and <50 mg/dL in females | <40 mg/dL in males and <50 mg/dL in females, or specific treatment for this lipid abnormality. | <40 mg/dL in males and <50 mg/dL in females, or specific treatment for this lipid abnormality. |
| **High blood pressure** | SBP≥130 mmHg and/or DBP≥85mmHg | SBP≥130 and/or DBP≥85mmHg, or treatment of previously diagnosed hypertension. | SBP≥130 mmHg and/or DBP≥85mmHg |
| **Raised fasting blood glucose** | ≥110 mg/dL. | ≥100mg/dl or previously diagnosed type 2 diabetes mellitus. | ≥100mg/dl or previously diagnosed type 2 diabetes mellitus. |
| **Criteria** | Meeting at least three out of five criteria | Waist circumference expanding plus any two of the other criteria | Meeting at least three out of five criteria |

Abbreviations: NCEP ATP III, National Cholesterol Education Program's Adult Treatment Panel III diagnostic criteria for MS; IDF, New International Diabetes Federation diagnostic criteria for MS; Harmonized criteria, modified criteria proposed by IDF, National Heart, Lung and Blood Institute, American Heart Association, World Heart Federation, International Atherosclerosis Society and International Association for the Study of Obesity.

extracted data from the included studies and discussed with a third one of us (XW) when there was any disagreement. The gathered information included author and publication year, study type, country, diagnostic criteria of MS, sample size, prevalence of MS, sex and mean age of both groups.

## Quality of the included studies

The quality assessment of the observational studies was performed based on the Newcastle-Ottawa Scale (NOS) [17]. The quality assessment tool was used to evaluate the validity of the included studies on three broad perspectives: the selection of the study groups; the comparability of the groups; and the ascertainment of either the exposure or outcome of interest. A study with a score ≥7 was considered high quality. Two authors (JY and WX) independently assessed the quality of each original study using the tool. Disagreements were resolved through discussion.

## Statistical analysis

Review Manager 5.3 software (freeware available from The Cochrane Collaboration) was used to statistically analyze the data. We calculated the pooled odds ratios (ORs) and 95% confidence intervals (CIs) of the prevalence of MS in both groups using dichotomous data that were either provided by the studies or calculated data from the studies. The heterogeneity among the included studies was assessed by p and $I^2$. When there was no heterogeneity (p>0.1, $I^2$≤50%), a Mantel-Haenszel (M-H) fixed-effect model was used; whereas a random-effect model was used if heterogeneity among studies was significant (p≤0.1, $I^2$>50%). A value of P<0.05 was considered statistically significant. Subgroup analysis of studies with different diagnostic criteria was also performed. Publication bias was assessed via visual inspection with a funnel plot when there were more than 8 articles.

## Results

### Literature search

We searched 4 databases (PubMed, Embase, the Cochrane Library and Web of Science) using the search term "(metabolic syndrome OR syndrome X OR insulin resistance syndrome OR Reaven syndrome) AND (lichen planus OR lesion planus)", and it resulted 199 records. Duplicates, irrelevant articles, reviews, case reports were excluded after screening on titles and abstracts. After a further reading, 11 studies plus 1 additional record identified through articles and citations fulfilled the eligibility criteria, with a total of 1422 participants (715 with LP and 707 controls). Literature screening process and a corresponding flow chart are shown in Fig 1.

### Study characteristics

The literatures included were all clinic-based studies and presented 5 different countries (Table 2). 6 studied [18–23] were conducted in India, 3 [24–26] in Egypt, 1 [27] in Spain, 1 [28] in Turkey and 1 [29] in Nigeria. The population for the controls without LP were apparently healthy individuals or outpatients with skin diseases other than LP, psoriasis, atopic dermatitis, and vitiligo (mainly nevi, seborrheic keratosis, actinic keratosis, verruca vulgaris, or basal cell carcinoma). MS in 7 [19–21, 23, 27–29] of the included studies were diagnosed by NCEP ATP III criteria [14], 3 [18, 24, 26] were diagnosed by IDF criteria [15] and 2 [22, 25] by Harmonized criteria [16].

There were no statistically significant differences with regard to age or sex between two groups in all literatures. Participants in the 12 studies were all older than 15 years of age. The

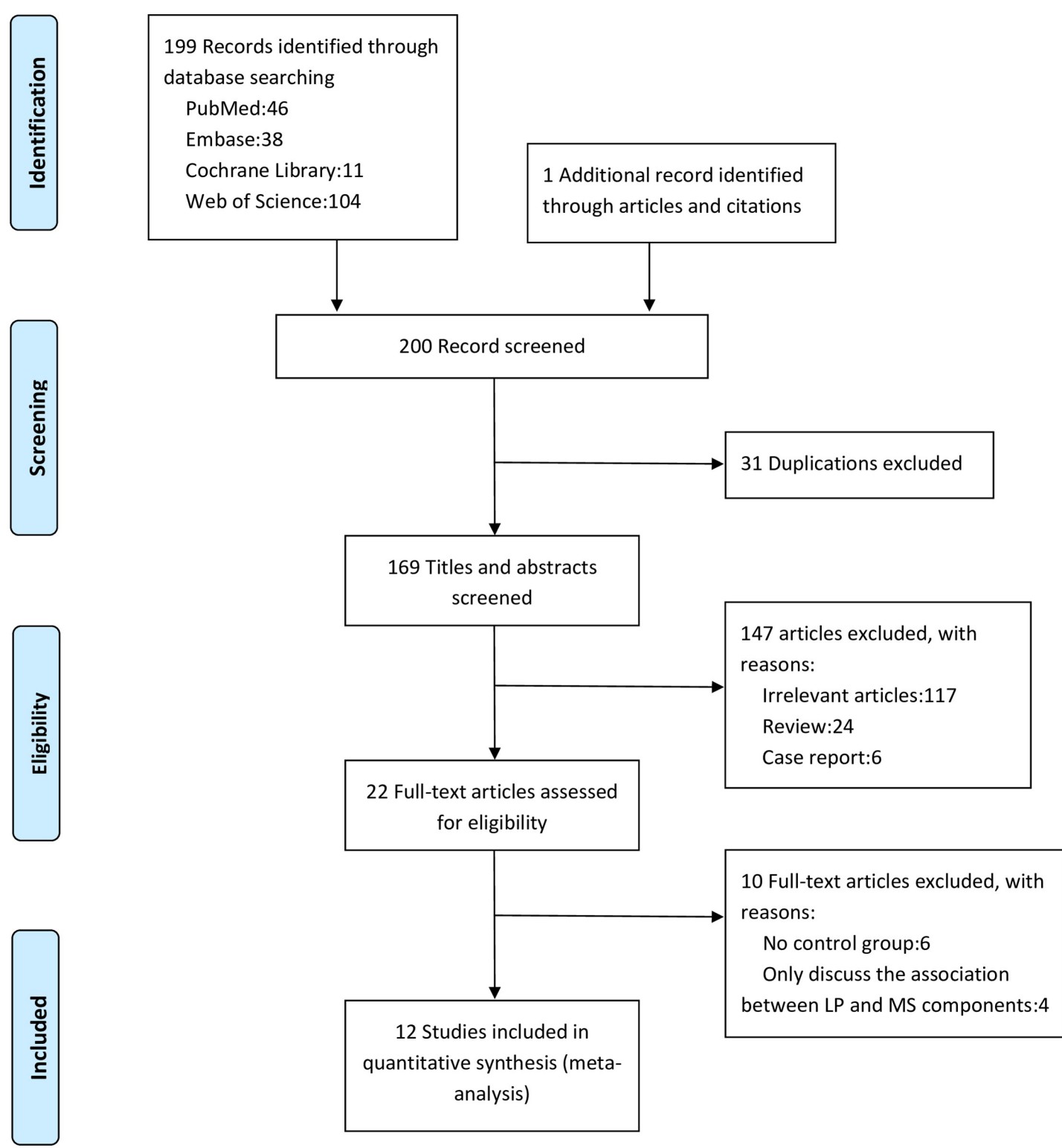

**Fig 1. The literature screening process.**

**Table 2. Characteristics of included studies in the meta-analysis and the participants enrolled.**

| Included studies | study type | country | MS Diagnostic Criteria | No. of participants | | No. of MS | | Mean age,y | | No. of M/F | |
|---|---|---|---|---|---|---|---|---|---|---|---|
| | | | | LP | C | LP | C | LP | C | LP | C |
| Agarwala et al., 2016 [18] | Abstract | Indian | IDF | 39 | 78 | 23 | 25 | NR | NR | NR | NR |
| Al Refai et al., 2018 [24] | Case-control study | Egypt | IDF | 40 | 40 | 22 | 6 | 38.1±11.8 | 38±9 | 20/20 | 20/20 |
| Arias-santiago et al., 2011 [27] | Case-control study | Spain | NCEP-ATP III | 100 | 100 | 27 | 20 | 47.4±9 | 48.3±7 | 50/50 | 50/50 |
| Baykal et al., 2015 [28] | Case-control study | Turkey | NCEP-ATP III | 79 | 79 | 21 | 10 | 47.11±14.44 | 46.9±14.32 | 30/49 | 28/51 |
| Krishnamoorthy et al., 2014 [19] | Case-control study | Indian | NCEP-ATP III | 11 | 14 | 3 | 2 | 33.09 | 33.09 | NR | NR |
| Kumar et al., 2019 [20] | Case-control study | Indian | NCEP-ATP III | 75 | 75 | 13 | 11 | 35.12±8.03 | NR | 32/43 | NR |
| Kuntoji et al., 2016 [21] | Case-control study | Indian | NCEP-ATP III | 50 | 50 | 3 | 1 | 41.24±16.17 | 39.48±11.36 | 28/22 | 29/21 |
| Mushtaq et al., 2020 [22] | Cross-sectional study | Indian | Harmonized criterion | 61 | 61 | 18 | 6 | 42.48±13.47 | 42.46±13.02 | 36/25 | 36/25 |
| Okpala et al., 2019 [29] | Cross-sectional study | Nigeria | NCEP-ATP III | 90 | 90 | 17 | 12 | 37.44±13.88 | 37.47±12.24 | 48/42 | 42/58 |
| Saleh et al., 2014 [25] | Case-control, study | Egypt | Harmonized criterion | 40 | 40 | 31 | 0 | 38.2±11.8 | 33.1±9.6 | 20/20 | 20/20 |
| Sharaf et al., 2017 [26] | Case-control study | Egypt | IDF | 30 | 30 | 14 | 3 | 36 ± 11.52 | 36 ± 10.65 | 24/6 | 24/6 |
| Singla et al., 2019 [23] | Cross-sectional study | Indian | NCEP-ATP III | 100 | 50 | 43 | 13 | 42.02 ± 13.82 | 40.72 ± 10.83 | 60/40 | NR |

Abbreviations: MS, metabolic syndrome; LP, lichen planus group; C, control group; NR, not reported; M/F, male/female.

studies included 1 abstract [18], 3 cross-sectional studies [22, 23, 29] and 8 case-control studies [19–21, 24–28], which characteristics are detailed in Table 2.

## Quality of included studies

All the included studies were rated with a score according to the NOS guidelines. Eleven studies were rated "good quality" with 7 or more stars, indicating a low risk of bias. One study was rated "adequate quality" with 6 stars, indicating a high risk of bias. The rating details are provided in Table 3.

## Meta-analysis results

We identified MS in the LP and control groups in 12 studies and observed marked heterogeneity across these studies (p = 0.009, $I^2$ = 56%); therefore, the M-H random-effect model was used. Overall, the prevalence of MS was significantly higher in the LP group than in the control group (odds ratio [OR] 2.81, 95% confidence interval [CI] 1.79–4.41, p<0.00001, Fig 2).

We also performed subgroup analysis of the prevalence of MS diagnosed by the NCEP ATP III criteria in 7 observational studies, the IDF criteria in 3 studies, and the Harmonized criteria in 2 studies separately. The pooled OR was much higher for studies that used the IDF diagnostic criteria (OR 4.65, 95% CI 2.52–8.58, p<0.00001, Fig 3) and the Harmonized criteria (OR 26.62, 95% CI 0.29–2471.37, p = 0.16, Fig 3) than for studies using the NCEP ATP III criteria (OR 1.75, 95% CI 1.25–2.44, p = 0.001, Fig 3), suggesting that studies using the IDF criteria and the Harmonized criteria reported a stronger association between MS and LP.

## Risk of bias assessment

Considering potential publication bias, we generated a funnel plot of ORs on the x-axis against the standard error (logOR) of each study on the y-axis (Fig 4). Among the 12 studies included in this meta-analysis, the scatter funnel plot with estimable ORs appeared symmetrical, indicating the absence of publication bias.

**Table 3. Quality of included studies.**

| Included studies | Selection | | | | Compatibility | Exposure | | | Total stars |
|---|---|---|---|---|---|---|---|---|---|
| | Is the Case Definition Adequate? | Representativeness of the Case | Selection of Controls | Definition of Controls | | Ascertainment of Exposure | Same method of ascertainment for cases and controls | Nonresponse Rate | |
| Agarwala et al., 2016 [18] | ※ | ※ | | | ※※ | ※ | ※ | ※ | 7 |
| Al Refai et al., 2018 [24] | ※ | ※ | | ※ | ※※ | ※ | ※ | ※ | 8 |
| Arias-santiago et al., 2011 [27] | ※ | ※ | | ※ | ※※ | ※ | ※ | ※ | 8 |
| Baykal et al., 2015 [28] | ※ | ※ | | ※ | ※※ | ※ | ※ | ※ | 8 |
| Krishnamoorthy et al., 2014 [19] | ※ | ※ | | ※ | ※※ | ※ | ※ | ※ | 8 |
| Kumar et al., 2019 [20] | ※ | ※ | | ※ | ※※ | ※ | ※ | ※ | 8 |
| Kuntoji et al., 2016 [21] | | ※ | | ※ | ※※ | ※ | ※ | ※ | 7 |
| Mushtaq et al., 2020 [22] | ※ | ※ | | ※ | ※※ | ※ | ※ | ※ | 8 |
| Okpala et al., 2019 [29] | ※ | ※ | | ※ | ※※ | ※ | ※ | ※ | 8 |
| Saleh et al., 2014 [25] | ※ | ※ | | ※ | ※※ | ※ | ※ | ※ | 8 |
| Sharaf et al., 2017 [26] | | ※ | | ※ | ※※ | ※ | ※ | ※ | 7 |
| Singla et al., 2019 [23] | | ※ | | | ※※ | ※ | ※ | ※ | 6 |

## Discussion

To our knowledge, this is the first systematic review and meta-analysis of studies that investigated the prevalence of MS in patients with LP compared to the general population. This

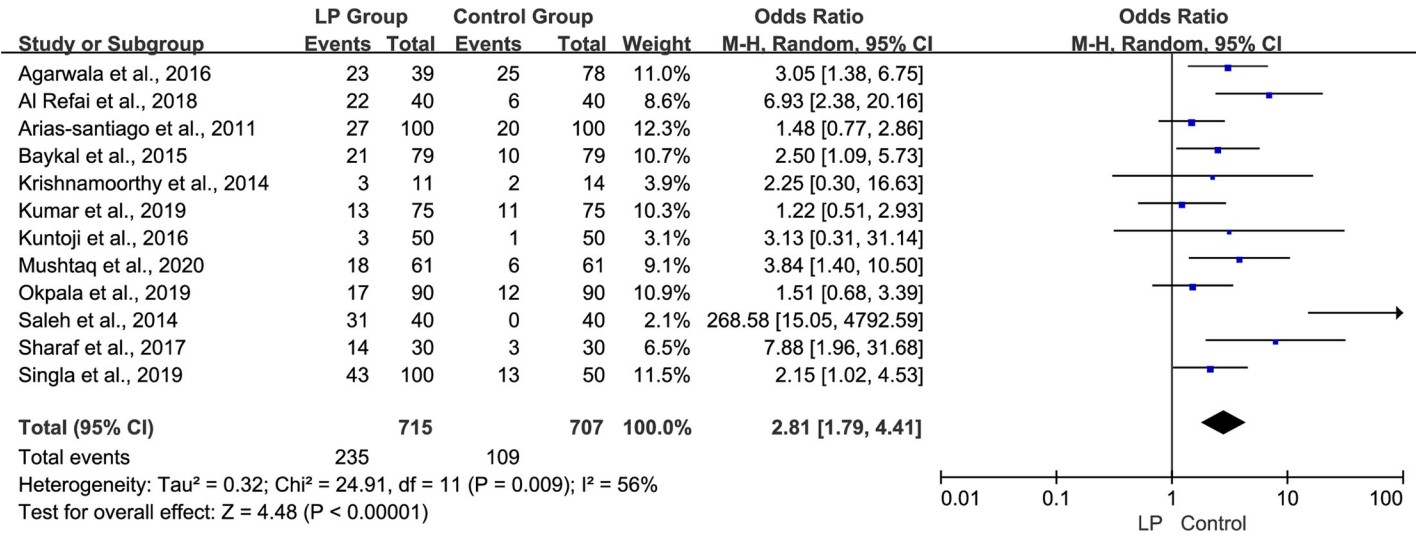

**Fig 2. Forest plot of the overall prevalence of MS in LP patients in the observational studies.** Abbreviations: M-H, a Mantel-Haenszel model; CI, confidence interval.

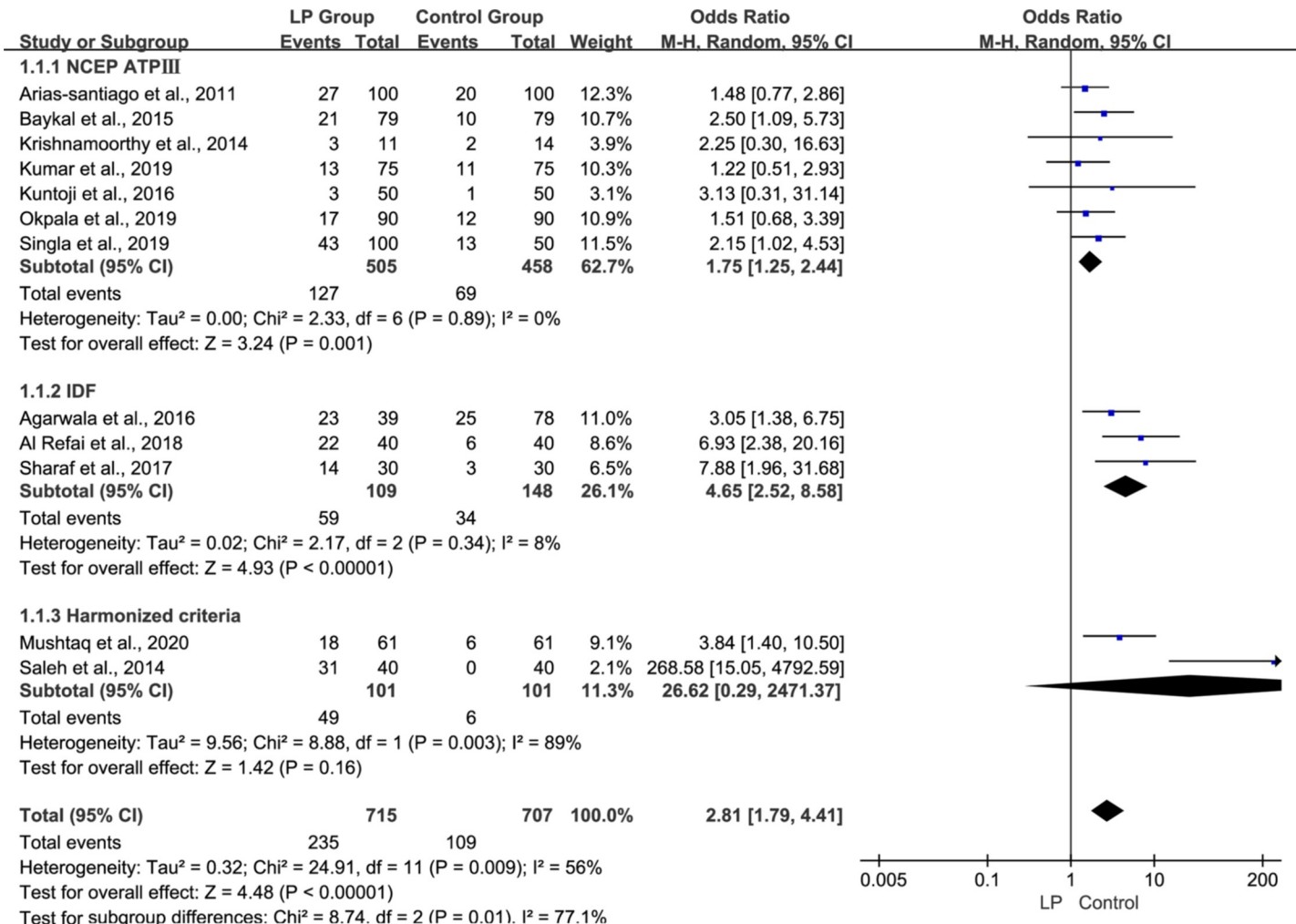

**Fig 3. Forest plot of subgroup analysis of MS diagnosed by different diagnostic criteria in the observational studies.**

analysis shows that compared with the general population, patients with LP showed a statistically significant approximately 2-fold higher prevalence of MS, which was present in 32.87% of LP patients versus 15.42% of the controls.

The high risk of MS in patients with LP may be attributable to chronic inflammation observed in this patient population. T cell activation in LP triggers the release of pro-inflammatory cytokines such as IL-2, IL-4, IL-6, IL-10, interferon-gamma, and TNF-α, which promote the release of more cytokines. These cytokines also play an important role in the development of dyslipidemia [30]. Some researchers are of the view that chronic inflammatory markers are useful predictors of cardiovascular events [31]. Notably, oxidative stress injury is implicated as an important pathogenetic contributor to MS. Mitran et al. reported increased lipid peroxidation and impairment of the antioxidant defense mechanism in patients with LP [32].

Current studies reported an association between LP and cardiovascular disease. Sahin et al. reported increased P-wave dispersion in patients with LP [33]. A study performed by Baykal et al. reported higher systolic and diastolic blood pressures and arterial stiffness in patients with LP [34]. Notably, 10 [18, 20–27, 29] studies from the available literature, which were

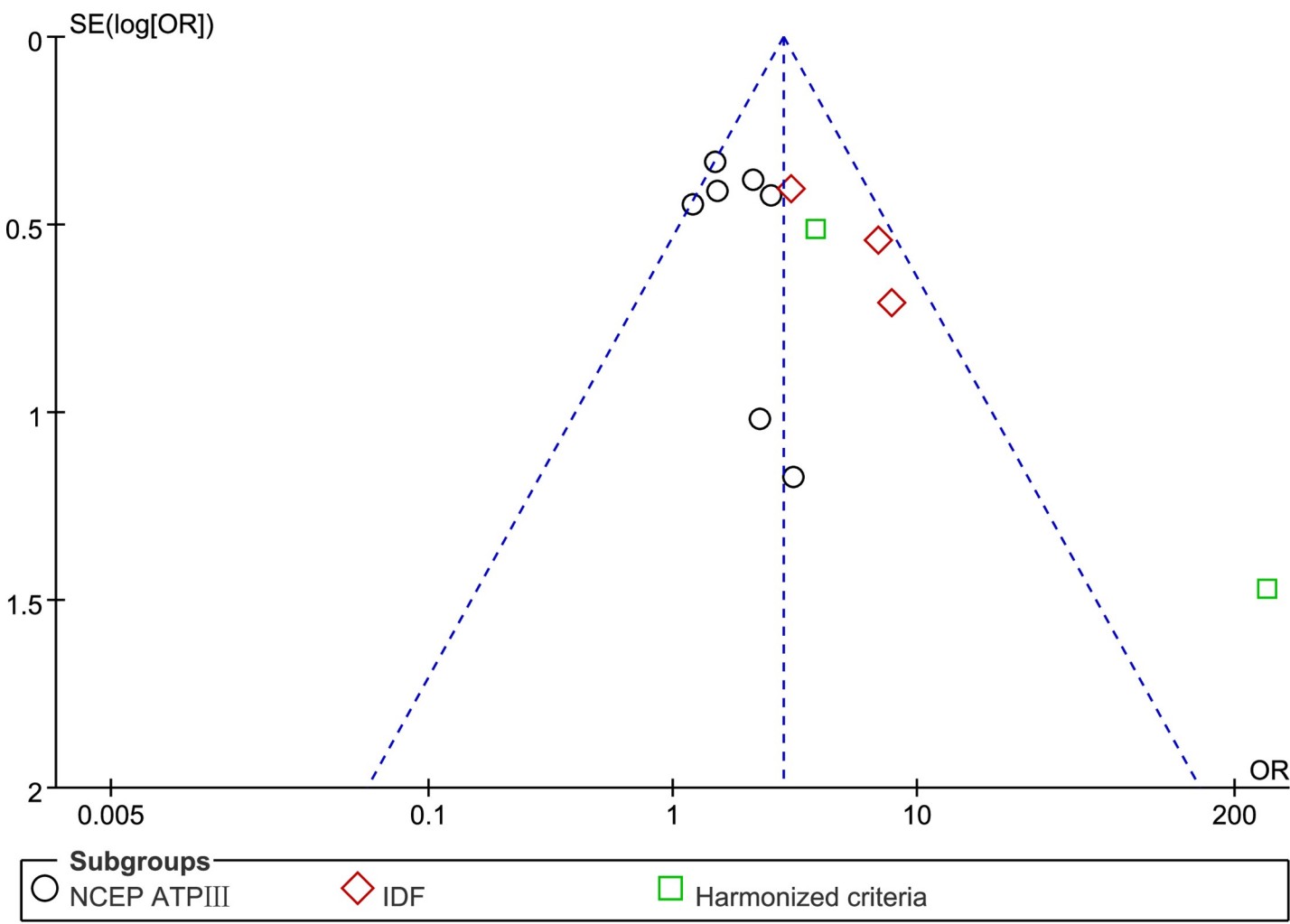

**Fig 4. Funnel plot of the overall prevalence of MS in LP patients in 12 observational studies.** Abbreviations: OR, odds ratio; SE, standard error.

included in this meta-analysis, reported a high prevalence of dyslipidemia in patients with LP. Another meta-analysis by Lai et al. also reported the same findings; the authors observed that LP was significantly associated with a high risk of dyslipidemia (specifically, high TG levels) [35].

The risk of MS varies depending on the clinical types of LP observed in patients. Baykal et al. observed that patients with specifically mucosal LP showed a higher prevalence of MS [28]. Kumar et al. reported that compared with the overall prevalence of LP, oral LP was more common in patients with hypothyroidism [20]. Okpala et al. observed that patients with and without dyslipidemia showed a higher likelihood of developing hypertrophic and classic LP, respectively [29].

Subgroup analysis of the different diagnostic criteria of MS showed that the pooled OR was much higher for studies that used the IDF diagnostic criteria (OR 4.65, 95% CI 2.52–8.58, p<0.00001, Fig 3) and the Harmonized criteria (OR 26.62, 95% CI 0.29–2471.37, p = 0.16) than for studies using the NCEP ATP III criteria (OR 1.75, 95% CI 1.25–2.44, p = 0.001, Fig 3), suggesting that studies using the IDF diagnostic criteria and the Harmonized criteria reported a stronger association between MS and LP. This finding could be attributed to the lower

obesity thresholds used to define central obesity, which consider a waist circumference of at least 94 cm in men and 80 cm in women. Therefore, based on this criterion, a greater number of patients would be diagnosed with MS. Some researchers are of the view that MS prevalence is higher using the IDF criteria; therefore, this definition could be more appropriate to diagnose MS [36]. Subramani et al. recorded maximum prevalence of MS when the Harmonized criteria was followed, and observed good agreement between Harmonized criteria and IDF criteria [37]. In this meta-analysis, we could not definitively conclude whether Harmonized criteria is better than IDF diagnostic criteria because only a few relevant articles were available for analysis.

### Limitations

Following are the limitations of our meta-analysis: (a) We observed a high degree of variability across studies owing to heterogeneity ($I^2$ = 56%, Fig 2). Patients enrolled in this study included those with different types, severity levels, and courses of LP, as well as varying duration of follow-up. Moreover, the diagnostic criteria for MS varied widely across studies. (b) The studies included in this meta-analysis were mainly performed in Asia, Africa, and Europe; therefore, these study samples might not be representative of the entire population. (c) Only a small number of studies were included in this meta-analysis; therefore, the low statistical power of this study is a drawback of this research. Considering the limitations of the current study, further prospective studies and high-quality research are warranted to definitively establish the association between MS and LP.

### Conclusions

This meta-analysis shows that compared with the general population, patients with LP are more likely to develop MS. Therefore, early diagnosis and prompt initiation of first-line therapy for metabolic disorders are important in patients with LP.

### Supporting information

**S1 Checklist. PRISMA checklist.** PRISMA statement for reporting systematic reviews and meta-analyses.
(DOC)

**S1 Table. Search strategy.**
(DOCX)

### Acknowledgments

We would like to thank Ms. Yuan Zu and Ms. Yujia Cai for their assistance with the statistical aspects of this meta-analysis.

### Author Contributions

**Conceptualization:** Jieya Ying, Wenzhong Xiang, Yu Qiu, Xiaofang Zeng.

**Data curation:** Jieya Ying, Wenzhong Xiang, Yu Qiu, Xiaofang Zeng.

**Formal analysis:** Jieya Ying, Wenzhong Xiang, Yu Qiu, Xiaofang Zeng.

**Funding acquisition:** Wenzhong Xiang, Xiaofang Zeng.

**Investigation:** Jieya Ying, Yu Qiu.

**Methodology:** Jieya Ying, Xiaofang Zeng.

**Project administration:** Wenzhong Xiang.

**Software:** Jieya Ying, Yu Qiu, Xiaofang Zeng.

**Supervision:** Wenzhong Xiang, Xiaofang Zeng.

**Validation:** Wenzhong Xiang, Yu Qiu, Xiaofang Zeng.

**Visualization:** Wenzhong Xiang, Xiaofang Zeng.

**Writing – original draft:** Jieya Ying, Yu Qiu.

**Writing – review & editing:** Jieya Ying, Yu Qiu.

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
