## [Decision Letter · Decision Letter 0]

25 Jun 2020

PONE-D-20-09664

Risk of metabolic syndrome in patients with lichen planus : A Systematic Review and Meta-analysis

PLOS ONE

Dear Dr. Xiang,

Thank you for submitting your manuscript to PLOS ONE. After careful consideration, we feel that it has merit but does not fully meet PLOS ONE’s publication criteria as it currently stands. Therefore, we invite you to submit a revised version of the manuscript that addresses the points raised during the review process.

We look forward to receiving your revised manuscript.

Kind regards,

Beatrice Nardone

Academic Editor

PLOS ONE

Journal Requirements:

2. At this time we ask that you provide the following additional information in the Methods section of you manuscript:

1.    Please provide the complete search strategy for at least one database as a new supporting information file.

2.    In the Methods, please specify how study quality was assessed.

3.    Please provide the date on which the search of databases was completed

Additional Editor Comments (if provided):

Please, carefully address the reviewers' comments. Also, please explain the protocol used for this review if any.

Reviewers' comments:

Reviewer's Responses to Questions

**Comments to the Author**

1. Is the manuscript technically sound, and do the data support the conclusions?

Reviewer #1: Partly

Reviewer #2: Yes

Reviewer #3: Yes

2. Has the statistical analysis been performed appropriately and rigorously? 

Reviewer #1: I Don't Know

Reviewer #2: Yes

Reviewer #3: Yes

3. Have the authors made all data underlying the findings in their manuscript fully available?

Reviewer #1: Yes

Reviewer #2: Yes

Reviewer #3: Yes

4. Is the manuscript presented in an intelligible fashion and written in standard English?

Reviewer #1: Yes

Reviewer #2: No

Reviewer #3: No

5. Review Comments to the Author

Reviewer #1: Major comments:

Why did the authors did not differ between the various clinical forms of LP?

What is new about the findings compared to previously published data since link between LP and MS hat been previously reported?

Reviewer #2: Overall meta-analysis is interesting and well prepared according to PRISMA. Selection of articles and applied methods are performed correctly. The results are important and may have practical application on lichen planus patients management.

I have some comments and suggestions:

1. The sentence in the Introduction: “MS is a complex group of metabolic disorders, which include central obesity, dyslipidemia, hypertension, and hyperglycemia. MS is implicated as an important risk factor for type 2 diabetes mellitus (T2DM) and cardiovascular disease (CVD) and has therefore attracted increasing attention in recent years.” is repeted in the Disscussion: “MS comprises a combination of metabolic disorders including central obesity, dyslipidemia, hypertension, and hyperglycemia that predispose individuals to T2DM and CVD.”

2. In the first sentence in Introduction it is better use the word “flat” instead of “plan”. The sentence: “Studies have indicated that insulin resistance[4], 3 oxidative stress injury[5] and chronic inflammatory[6] play important roles in the pathogenesis of MS” need to be corrected. Others grammar mistakes should be also corrected.

“…a similar chronic inflammatory dermatological condition with an autoimmune etiology, such as psoriasis[7] alopecia areata[8], hidradenitis suppurativa[9] and vitiligo[10]….”

3. The sentence: “T cell activation in LP triggers the release of pro-inflammatory cytokines such as IL-2, IL-4, IL-6, IL10, interferon-gamma, and TNF-α, which promote the release of more cytokines that attack keratinocytes resulting in dyslipidemia[29]” is unclear and should be rewrite.

4. “To our knowledge, this is the first systematic review and meta-analysis of studies that investigated the prevalence of MS in both patients with LP and the general population.” – what was the aim of the study, why authors have stated that it is the first study investigated the prevalence of MS in the general population?

5. In the Discussion the sentence: “Lipid abnormalities are significant contributors to the onset and aggravation of CVD. 11 The accumulation of cholesterol in cells and formation of lipid-laden foam cells produces fatty streaks in arterial walls, which predispose an individual to atherosclerotic plaques and consequent CVD[29]”

- the information is obvious and well known to readers of the journal and should be removed.

Reviewer #3: This article gives an interesting perspective on a little studied topic. It is a meta-analysis to analyze the association between metabolic syndrome and lichen planus and shows that compared with the general population, patients with lichen planus are more likely to develop metabolic syndrome. The work is a valuable contribution to the Plos One readers. However, it needs corrections, as suggested to the authors.

Materials and methods

Page 4

Data sources and searches

As the terms syndrome x, insulin resistance syndrome and Reaven syndrome were used for the search, specify these synonyms in the introduction.

Inclusion criteria

Outcome measures: "MS was diagnosed using the National Cholesterol Education Program Adult Treatment Panel III (NCEP ATP III), the International Diabetes Federation (IDF) and other diagnostic criteria."

Cite the references for the criteria or define what they are.

Exclusion criteria

....(2) For all participants: a known diagnosis of oral lichenoid reactions (a drug-induced LP-like reaction), hypertension, diabetes, dyslipidemia, chronic liver disease, chronic kidney disease, human immunodeficiency virus infection, and hereditary diseases.

Why were participants with hypertension, diabetes and dyslipidemia excluded if these characteristics are criteria for the diagnosis of metabolic syndrome?

Page 6.

Literature search

"After a further reading, 11 studies with 1300 participants fulfilled the eligibility criteria".

Specify how many patients were in the lichen planus group and how many were controls.

Study characteristics

"The population for the controls without LP were apparently healthy individuals or outpatients with or without skin diseases other than LP".

There are diseases that are associated with metabolic syndrome. Were patients with these diseases excluded?

Results

Table 2, figures 2 and 3

In included studies complete with "et al." after the first author in studies that have more than one author.

Discussion

Page 10

First paragraph: "To our knowledge, this is the first systematic review and meta-analysis of studies that investigated the prevalence of MS in both patients with LP and the general population".

Change to: "To our knowledge, this is the first systematic review and meta-analysis of studies that investigated the prevalence of MS in patients with LP compared to the general population".

The authors should also include bibliographic references that mention the frequency of metabolic syndrome in the general population at the end of the paragraph.

Second paragraph: "...T cell activation in LP triggers the release of pro-inflammatory cytokines such as IL-2, IL-4, IL-6, IL- 10, interferon-gamma, and TNF-α, which promote the release of more cytokines that attack keratinocytes resulting in dyslipidemia".

It is not the attack on keratinocytes that results in dyslipidemia.

It would be more accurate to say: "These cytokines also play an important role in the development of dyslipidemia".

References

The references are not within the journal's formatting standards

The manuscript needs editing for language quality.

6. PLOS authors have the option to publish the peer review history of their article (what does this mean?). If published, this will include your full peer review and any attached files.

Reviewer #1: No

Reviewer #2: No

Reviewer #3: No

---

## [Author Response · Author response to Decision Letter 0]

21 Jul 2020

Dear Beatrice Nardone and reviewers:

Thank you for your letter and the reviewers’ comments on our manuscript entitled “Risk of metabolic syndrome in patients with lichen planus: A systematic review and meta-analysis” (ID: PONE-D-20-09664). Those comments are very helpful for revising and improving our paper, as well as the important guiding significance to other research. We have studied the comments carefully and made corrections which we hope meet with approval. The main corrections are in the manuscript and the responds to the reviewers’ comments are as follows.

Replies to the reviewers’ comments:

Reviewer #1:

1. Why did the authors did not differ between the various clinical forms of LP?

Response：That's an excellent question. Clinical forms of LP including linear LP, hypertrophic LP, oral LP, LP pigmentosus, generalized LP, lichen planopilaris, bullous LP and actinic LP. Emerging evidences suggest that risk of MS varies depending on the clinical types of LP. However, only 3 out of 12 included studies provided original data or assessed the association between MS and different clinical forms of LP. Moreover, the amount of data after classification is too small to draw any conclusions. It is a prospective direction for further research work with research value, and we are planning to take this as the next research direction. 

2. What is new about the findings compared to previously published data since link between LP and MS hat been previously reported?

Response: Previous studies investigating the relationship between LP and MS were observational studies, and no meta-analysis in this area has yet been published. In this meta-analysis, we examined data on the relationship between LP and odds of MS by searching for studies published before July 16, 2020. In addition, we also conducted a comparison of three different diagnostic criteria of MS.

Reviewer #2: 

1. what was the aim of the study, why authors have stated that it is the first study investigated the prevalence of MS in the general population? 

Response：The aim of the study is to systematically evaluate the published literatures and determine a clinical relationship between MS and LP. Currently, no systematic review and meta-analysis in this area has yet been published; therefore, we stated that “this is the first systematic review and meta-analysis of studies that investigated the prevalence of MS in patients with LP compared to the general population.”

Reviewer #3:

1. Why were participants with hypertension, diabetes and dyslipidemia excluded if these characteristics are criteria for the diagnosis of metabolic syndrome?

Response：Firstly, we aim to determine whether patients with LP are more likely to develop MS when compared with the general population; therefore, we need untested LP patients and controls, who are not sure if they have had MS before. Besides, participants with known hypertension, diabetes and dyslipidemia often have a long history of medication, and we cannot rule out the possibility of drug-induced MS. 

2. There are diseases that are associated with metabolic syndrome. Were patients with these diseases excluded?

Response：Yes, skin disease such as psoriasis, atopic dermatitis, and vitiligo that are associated with MS were excluded. The outpatients with skin diseases were mainly nevi, seborrheic keratosis, actinic keratosis, verruca vulgaris, or basal cell carcinoma. We have added these to the “Materials and methods--Exclusion criteria” and “Results--Study characteristics” section.

3. We’ve also made the relevant changes to the Inclusion criteria, Literature search, Results and Discussion accordingly.

In addition, we have made the following changes to the article:

1. All three reviewers made reference to awkward sentences and lack of clarity in the flow of the text. We’ve addressed this issue across and within each section, paragraph, and sentence.

2. We re-searched the literature prior to July 16, 2020 according to the search strategy, and added 1 recently published eligible literature (Mushtaq S, Dogra D, Dogra N, Shapiro J, Fatema K, Faizi N, et al. Cardiovascular and Metabolic Risk Assessment in Patients with Lichen Planus: A Tertiary Care Hospital-based Study from Northern India. Indian Dermatol Online J. 2020;11(2):158-66.). 

3. We removed Medline from the databases because PubMed comprises citations for biomedical literature from MEDLINE. The number of results is different because the new PubMed, launched after 18 May 2020, has some changes in search syntax and search translations. 

4. We reedited the article to ensure that it meets PLOS ONE's style requirements and assessed study quality based on the Newcastle-Ottawa Scale. Complete search strategy is shown in S2 Table. 

Once again, thank you very much for your constructive comments and suggestions which would help us both in English and in depth to improve the quality of the paper.

Best wishes,

Wenzhong Xiang

E-mail: xiangwenzhong@126.com

---

## [Decision Letter · Decision Letter 1]

7 Aug 2020

Risk of metabolic syndrome in patients with lichen planus : A Systematic Review and Meta-analysis

PONE-D-20-09664R1

Dear Dr. Xiang,

We’re pleased to inform you that your manuscript has been judged scientifically suitable for publication and will be formally accepted for publication once it meets all outstanding technical requirements.

Kind regards,

Beatrice Nardone

Academic Editor

PLOS ONE

Additional Editor Comments (optional):

Reviewers' comments:

Reviewer's Responses to Questions

**Comments to the Author**

1. If the authors have adequately addressed your comments raised in a previous round of review and you feel that this manuscript is now acceptable for publication, you may indicate that here to bypass the “Comments to the Author” section, enter your conflict of interest statement in the “Confidential to Editor” section, and submit your "Accept" recommendation.

Reviewer #2: (No Response)

Reviewer #3: All comments have been addressed

2. Is the manuscript technically sound, and do the data support the conclusions?

Reviewer #2: Yes

Reviewer #3: Yes

3. Has the statistical analysis been performed appropriately and rigorously? 

Reviewer #2: Yes

Reviewer #3: Yes

4. Have the authors made all data underlying the findings in their manuscript fully available?

Reviewer #2: Yes

Reviewer #3: Yes

5. Is the manuscript presented in an intelligible fashion and written in standard English?

Reviewer #2: Yes

Reviewer #3: Yes

6. Review Comments to the Author

Reviewer #2: The results presented in the manuscript are interesting and may have some significant influence on the management of lichen planus patients in clinical practice. The manuscript has been corrected and improved. The article may be published in this form.

Reviewer #3: The authors have adequately addressed my comments raised in a previous round of review and I feel that this manuscript is now acceptable for publication.

7. PLOS authors have the option to publish the peer review history of their article (what does this mean?). If published, this will include your full peer review and any attached files.

Reviewer #2: No

Reviewer #3: **Yes: **Marilda Aparecida Milanez Morgado de Abreu

---

## [Editor Report · Acceptance letter]

11 Aug 2020

PONE-D-20-09664R1 

Risk of metabolic syndrome in patients with lichen planus: A Systematic Review and Meta-analysis 

Dear Dr. Xiang:

I'm pleased to inform you that your manuscript has been deemed suitable for publication in PLOS ONE. Congratulations! Your manuscript is now with our production department. 

Kind regards, 

on behalf of

Dr. Beatrice Nardone 

Academic Editor

PLOS ONE